# H-nobs: Achieving Certified Fairness and Robustness in Distributed Learning on Heterogeneous Datasets

**Guanqiang Zhou**
George Mason University
gzhou4@gmu.edu

**Ping Xu**
University of Texas Rio Grande Valley
ping.t.xu@utrgv.edu

**Yue Wang**
Georgia State University
ywang182@gsu.edu

**Zhi Tian**
George Mason University
ztian1@gmu.edu

## Abstract

Fairness and robustness are two important goals in the design of modern distributed learning systems. Despite a few prior works attempting to achieve both fairness and robustness, some key aspects of this direction remain underexplored. In this paper, we try to answer three largely unnoticed and unaddressed questions that are of paramount significance to this topic: (i) What makes jointly satisfying fairness and robustness difficult? (ii) Is it possible to establish theoretical guarantee for the dual property of fairness and robustness? (iii) How much does fairness have to sacrifice at the expense of robustness being incorporated into the system? To address these questions, we first identify data heterogeneity as the key difficulty of combining fairness and robustness. Accordingly, we propose a fair and robust framework called H-nobs which can offer certified fairness and robustness through the adoption of two key components, a fairness-promoting objective function and a simple robust aggregation scheme called norm-based screening (NBS). We explain in detail why NBS is the suitable scheme in our algorithm in contrast to other robust aggregation measures. In addition, we derive three convergence theorems for H-nobs in cases of the learning model being nonconvex, convex, and strongly convex, respectively, which provide theoretical guarantees for both fairness and robustness. Further, we empirically investigate the influence of the robust mechanism (NBS) on the fairness performance of H-nobs, the very first attempt of such exploration.

## 1   Introduction

Distributed learning has been widely applied to solve large-scale machine learning (ML) problems, where a number of computing nodes (users) collectively carry out the task of training a model under the coordination of a central node (server). Federated learning is a typical example of distributed learning where each device collects its own local data and keeps them private [1]. In this paper we use distributed learning and federated learning interchangeably.

Aside from the basic expectation of high model accuracy, it is increasingly emphasized in the latest ML research to design distributed learning algorithms that can jointly satisfy several pragmatic constraints such as fairness and robustness [2]. Here we consider the most common definitions where fairness specifically refers to egalitarian fairness and robustness refers to Byzantine robustness, which respectively stipulate that the trained model exhibits somewhat uniform performances on the local users (see Section 2.1), and that the system is resilient to Byzantine attacks where a portion of users defect or malfunction (see Section 2.2). In this paper, we strive for simultaneously

37th Conference on Neural Information Processing Systems (NeurIPS 2023).

attaining fairness and robustness. Although a few prior efforts have been made towards fair and robust distributed learning [3, 4, 5], this direction is largely underexplored, and some key issues have not been addressed yet. In this paper, we seek to answer three questions that are of paramount importance to the understanding of this topic.

The first question is "*What makes jointly satisfying fairness and robustness difficult?*" According to the recent literature, (un)fairness issues normally arise in the non-iid settings where the training data residing on local users are heterogeneous [6]. However, for a system to be Byzantine-robust, a typical underlying assumption is that the local data are homogeneous/iid [7, 8, 9, 10, 11]. Such a conflict would invalidate most of the existing Byzantine-robust methodologies developed under the iid assumptions. Therefore, to meaningfully achieve fairness and robustness at the same time, one has to address data heterogeneity, which entails both theoretical and empirical challenges illustrated as follows.

The second question is "*Is it possible to establish theoretical guarantee for the dual property of fairness and robustness?*" According to the literature, providing convergence guarantees for Byzantine-robust methods (alone) is challenging enough [12]. This is because these methods not only rely on the iid assumption, but also require strong assumptions on the distribution of local gradients, such as sub-exponential [9] or sub-Gaussian [10] (see Section 4.3 for more details). These distributional assumptions usually fall short of proper justifications, and they become even harder to justify under the modified objectives with fairness constraints. Due to these two major obstacles, there has not yet been any work that manages to provide theoretical guarantees for both fairness and robustness.

The third question is "*How much does fairness have to sacrifice at the cost of robustness?*" To display the inherent conflicts between these two goals, we consider the principles to achieve either goal separately. On one hand, fairness is typically enforced by up-weighting the local objective function of the worst-performing users, which would make their local gradients also heavily-weighted. On the other hand, robustness is typically enforced by the server discarding partial gradients that look suspicious. Since the server cannot decide whether the suspicious gradients (by having large norms or prominent entries) are sent by the Byzantine users or generated from the up-weighted objectives, the latter can be easily filtered out by the deployed robust mechanism, which would inevitably impair the fairness performance. Although this issue was first recognized by a prior work [4], such internal conflicts have not been studied, either theoretically or empirically. Consequently, there are no reported results or understanding on this matter.

**Our contributions.** In this paper, we propose a fair and robust framework called H-nobs, which contains two key components: a fairness-promoting objective function (the H part) and a simple robust aggregation scheme called norm-based screening/NBS (the nobs part). On one hand, in our algorithm, the certified fairness comes from the fairness-promoting objective function, which can be directly adopted from existing works with theoretical fairness guarantees (see Section 3.2 for a couple of representative examples). On the other hand, we show that the deployed NBS is able to gracefully handle Byzantine attacks and theoretically guarantee the convergence of our algorithm without requiring the iid assumption that is ubiquitous for convergence analysis. Accordingly, we derive three convergence theorems for H-nobs in cases of the learning model being nonconvex, convex, and strongly convex, respectively. In addition, we explain in great detail why NBS is distinctively suitable for our framework, and how it may not be the case for other robust aggregation measures. We achieve this by illustrating three notable theoretical advantages of NBS over its counterparts (see Section 4.3). To address the conflicts between fairness and robustness, we empirically investigate the influence of several robust aggregation schemes, including NBS and other widely-known benchmarks, on the fairness performance (see Section 5). Our results show that although the screening of some fairness-enhancing gradients is inevitable, the overall performance of H-nobs is satisfying. To the best of our knowledge, our work is the first that touches on all of the aforementioned aspects in the pursuit of fair and robust distributed learning.

The major contributions of this work can be summarized as follows:

- We recognize the major challenges in obtaining joint fairness and robustness in distributed learning by identifying three important yet largely overlooked questions. In this work, we have addressed these questions to varying degrees.
- We propose H-nobs, the first algorithm that takes the (obvious but challenging) route of incorporating robust aggregation into existing fair objectives to achieve fair and robust distributed learning.

- We achieve certified fairness and robustness for H-nobs even under the challenging non-iid scenarios, by establishing theoretical convergence guarantees with multiple levels of assumptions on the model convexity. Also, we clearly identify the major technical challenges that may hinder this goal, and illustrate how those challenges can be overcome by NBS through its theoretical robust property.

- We take the initiative to investigate the empirical conflicts between fairness and robustness under a specific metric and objective of fairness, and demonstrate the robustness and fairness benefits of H-nobs.

## 2 Related work

### 2.1 Fairness in federated learning

Differing from the common notion of fairness with respect to certain protected attributes (such as race and gender [13]), fairness in federated learning is typically defined over the model accuracies of local devices [14, 15], including two major fairness metrics, i.e., collaborative fairness [3] and egalitarian fairness [16]. These two metrics serve different purposes and suit for the corresponding scenarios. In particular, collaborative fairness enforces each user to receive a personalized model after each iteration according to its contribution to the training [3], which runs counter to our work assuming a single shared model for all users. Therefore, in this paper we only consider egalitarian fairness, which aims to achieve not only satisfactory overall model accuracy, but also somewhat uniform local performances for all participating users [16].

To enforce fairness, one common approach is to replace the empirical training loss with a modified objective function that puts a higher emphasis on the worst-performing users. For example, in an earlier work, [17] proposes Agnostic Federated Learning (AFL), a minimax optimization scheme to minimize the training loss of the worst-off client among all users. However, AFL is reported to perform well only on very small networks consisting of a handful of devices, and its generalization guarantees may not be applicable in a large-scale setting [14]. To improve upon AFL, [16] proposes $q$-Fair Federated Learning ($q$-FFL), a fairness-promoting objective, which enjoys better scalability and more favorable accuracy/fairness tradeoffs than AFL. The exact formulation of $q$-FFL is given in Section 3.2. In Section 5, we adopt egalitarian fairness as the fairness metric and $q$-FFL as the objective function to empirically evaluate the effectiveness of our framework.

### 2.2 Byzantine-robust techniques

Byzantine attack/failure refers to the issue where a small portion of failed nodes may send distorted or even adversarial messages to the server during training [18]. To cope with Byzantine attacks, there are generally two distinct approaches. The first approach assigns each user redundant data, and relies on this redundancy to eliminate the effects of erroneous messages [19, 12]. This approach may not be applicable to federated learning where data cannot be replicated and reassigned for user privacy. The second approach is based on certain robust aggregation mechanisms at the server's end, such as Krum [7], geometric median [8], coordinate-wise median [9], iterative filtering [10], etc. Due to its wide variety of aggregation rules and applicability to federated learning, robust aggregation is commonly viewed as the mainstream approach for mitigating Byzantine attacks.

Although it is an intuitive and seemingly straightforward idea to incorporate robust aggregation into the fairness-promoting objectives for one to achieve both fairness and robustness, we observe that few works take this route due to the two stringent assumptions mentioned in Introduction. To address these technical difficulties, we resort to norm-based screening (NBS), a low-profile screening scheme among the robust aggregation cohorts. Although NBS has previously been implemented in the Byzantine-prone settings [11, 20], few realize its effectiveness in the challenging non-iid environments and its handiness for convergence analysis without necessitating the stringent assumptions. In Section 4, we demonstrate that NBS enjoys a desirable property which allows it to guarantee convergence under much milder assumptions.

# 3 Proposed framework

## 3.1 Basic setting

**Data division.** We consider a typical distributed learning setup with one server and $m$ users, each holding $n$ data points with a total of $mn = N$ training samples. Note that we do not assume that the local data are iid, despite the same sample size for each user. Also, uneven data allocation cases with varying local sample sizes can easily fit into our framework with minor adjustments.

**Byzantine attack.** We assume that a percentage $\alpha$ of local users are Byzantine and the remaining $1 - \alpha$ are normal/honest. The sets of Byzantine users and honest users are denoted as $\mathcal{B}$ and $\mathcal{M}$ respectively, with $|\mathcal{B}| = \alpha m$ and $|\mathcal{M}| = (1-\alpha)m$. During each training iteration, the server would ask all users to conduct certain computational task based on their respective local data and to report the results back to the server. While honest users would follow the given instructions faithfully, Byzantine users need not to adhere to the protocol and can send arbitrary messages to the server.

## 3.2 Proposed algorithm

On the macro level, our algorithm is a variant of distributed gradient descent, characterized by two distinctive elements, i.e., fairness-aware objective and norm-based screening.

**Fairness-aware objective.** Let $f(\theta; x_j)$ be the point-wise training loss, we denote the local training loss on node $i$ as $F_i(\theta) = \frac{1}{n} \sum_{j=(i-1)n+1}^{(i-1)n+n} f(\theta; x_j)$ and the global training loss as $F(\theta) = \frac{1}{m} [F_1(\theta) + \cdots + F_m(\theta)]$. Since minimizing $F(\theta)$ can produce unfair models, we instead adopt a fairness-aware objective function $H(\theta)$ from existing literature which has a decomposable form $H(\theta) = \frac{1}{m} [H_1(\theta) + \cdots + H_m(\theta)]$, with $H_i(\theta)$ being the fairness-oriented version of $F_i(\theta)$ that solely relies on the local data of user $i$. A few such examples include:

- $H_i(\theta) = \frac{1}{n} \sum_{j=(i-1)n+1}^{(i-1)n+n} [f(\theta; x_j) - \mu]_+^2$ proposed by [6] where $\mu$ is a hyper-parameter.

- $H_i(\theta) = \frac{1}{q+1} [F_i(\theta)]^{q+1}$ (i.e., $q$-FFL) proposed by [16] where $q$ is a hyper-parameter.

Note that each specific $H_i(\theta)$ may aim to address a different notion of fairness, with the former example seeking to mitigate representation disparity between majority and minority groups and the latter promoting egalitarian fairness. We also want to point out that our work applies to a broad range of fairness-promoting objectives that are decomposable, thus not limited to the given two examples.

**Norm-based screening (NBS).** After the local gradients are computed by honest users (w.r.t. $H_i(\theta)$) and sent to the server (while Byzantine users craft disruptive gradients denoted as $\star$), we propose to use NBS to aggregate all local gradients before applying its output to update the model. As a robust aggregation measure, the idea of NBS is fairly simple: leave out the gradients with large norms and take the average of the remaining gradients as output. We formally define NBS as a function "**Norm_Screen**", as detailed in Algorithm 1.

---

**Algorithm 1** Norm-Based Screening

---

**Input:** $g_1, \ldots, g_m$ ($m$ local gradients), screening percentage $\beta$
**Output:** $G = \textbf{Norm\_Screen}_\beta(g_1, \ldots, g_m)$
  1: generate a new set of indices $(1), \ldots, (m)$, such that $\|g_{(1)}\| \leq \cdots \leq \|g_{(m)}\|$
  2: define an index set $\mathcal{U} = \{(1), \ldots, ((1-\beta)m)\}$, which specifies the unscreened gradients
  3: calculate the output by averaging the unscreened gradients $G = \frac{1}{|\mathcal{U}|} \sum_{i \in \mathcal{U}} g_i$

---

Despite its simplicity, NBS can gracefully handle Byzantine users, because any faulty gradient either is filtered out with a large norm, or can only shift the output to a bounded degree. In this way, if some Byzantine gradients bypass NBS, their influence on the output can be offset or mitigated if they are outnumbered by honest gradients that also remain unscreened. In Section 4, this property of NBS is formally stated as an inequality, which will serve as the bedrock of our convergence proofs later on. Combining the above two elements, we summarize the proposed algorithm in Algorithm 2.

**Algorithm 2** H-nobs: Fair & Byzantine-Robust Distributed Gradient Descent

---

**Input:** screening percentage $\beta$ ($\geq \alpha$), learning rate $\eta$, model initialization $\theta_0$, total iteration $T$
**Output:** completed model $\theta_T$
 1: **for** $t = 0, 1, \ldots, T-1$ **do**
 2:    **Server**: send $\theta_t$ to all users
 3:    **for** $i = 1, 2, \ldots, m$ **do**
 4:       **User** $i$: receive model $\theta_t$ from the server
 5:       calculate $\nabla H_i(\theta_t)$ according to the chosen form of $H_i(\theta_t)$
 6:       generate $g_i(\theta_t) = \begin{cases} \nabla H_i(\theta_t) & i \in \mathcal{M} \\ \star & i \in \mathcal{B} \end{cases}$
 7:       send $g_i(\theta_t)$ to the server
 8:    **end for**
 9:    **Server**: collect $g_1(\theta_t), \ldots, g_m(\theta_t)$ from the users
10:    compute the aggregated gradient $G(\theta_t) = \mathbf{Norm\_Screen}_\beta(g_1(\theta_t), \ldots, g_m(\theta_t))$
11:    update model $\theta_{t+1} = \theta_t - \eta \cdot G(\theta_t)$
12: **end for**

---

# 4 Convergence analysis

In this section, we begin by establishing a key theoretical property of NBS, and then develop three main convergence theorems, and conclude with a discussion that concretely states the distinctive advantages of NBS over other robust aggregation methods.

## 4.1 Preliminaries

Previously we explain how NBS can properly upper-bound the influence of Byzantine users in an intuitive way. Here we theoretically quantify this bound in Lemma 1, whose proof is deferred to Appendix A.

**Lemma 1.** *Suppose that a percentage of $\alpha \leq \frac{1}{2}$ among $m$ local gradients $g_1, \ldots, g_m$ are Byzantine, whose indices compose a set $\mathcal{B}$ ($|\mathcal{B}| = \alpha m$), and the index set of honest gradients is denoted as $\mathcal{M}$. With $G = \mathbf{Norm\_Screen}_\beta(g_1, \ldots, g_m)$ and $\beta \geq \alpha$, the following inequality holds:*

$$\|G - \nabla H\| \leq \frac{2\alpha}{1 - \beta}\|\nabla H\| + \max_{i \in \mathcal{M}} \|g_i - \nabla H\| \tag{1}$$

*where $\nabla H = \frac{1}{m}\big[\nabla H_1 + \cdots + \nabla H_m\big]$ with $\nabla H_i$ being the honest gradient of user $i$.*

The inequality (1) is critical to the convergence analysis. In Algorithm 2 the model is refined using $G$, which is the sum of two terms $\nabla H$ and $G - \nabla H$, with the former being the true gradient (noise-free) and the latter being a noise term that can take any direction. To prove convergence, we must upper-bound the norm of $G - \nabla H$ as (1) does.

To enable further analysis, we adopt Assumption 1 to bound the second term in the RHS of (1).

**Assumption 1.** *For any specific $\theta_t$, it always holds that*

$$\max_{1 \leq i \leq m} \left\| \nabla H_i(\theta_t) - \frac{1}{m}\sum_{k=1}^{m} \nabla H_k(\theta_t) \right\| \leq \sigma. \tag{2}$$

Assumption 1 obviously stands due to the simple fact that the distance between any single vector and the vector mean is always upper-bounded.

Moreover, we make Assumption 2 to ensure that the gradient $\nabla H(\theta)$ exists.

**Assumption 2.** *The fairness-aware objective $H(\theta)$ is $L_H$-smooth.*

## 4.2 Main theorems

**Nonconvex losses.** We first consider the most general case of the objective function $H(\theta)$ being nonconvex in $\theta$. For this case, we derive Theorem 1 that guarantees convergence of our algorithm to a stationary point of $H(\theta)$. The proof of Theorem 1 is deferred to Appendix B.

**Theorem 1.** *Suppose that Assumptions 1-2 hold and $\alpha < \frac{1}{3}$. Taking $\eta = \frac{1}{L_H}$, Algorithm 2 satisfies*

$$\frac{1}{T}\sum_{t=0}^{T-1} \|\nabla H(\theta_t)\|^2 \leq \frac{2L_H}{\left(1 - (1+r)C_\alpha^2\right)T}\left[H(\theta_0) - H(\theta^*)\right] + \frac{(1+1/r)\sigma^2}{1 - (1+r)C_\alpha^2} \tag{3}$$

*where $\theta^*$ is the global minimizer of $H(\theta)$, $C_\alpha = \frac{2\alpha}{1-\beta}$, and $r$ should satisfy $0 < r < \left(\frac{1-\beta}{2\alpha}\right)^2 - 1$.*

**Convex losses.** Now we consider the case where the loss function is convex as in Assumption 3. Additionally, we make Assumption 4 which suggests that all the intermediate iterations would not be infinitely worse than the initialization $\theta_0$. We derive Theorem 2 that grants our algorithm convergence guarantee in the convex regime. The proof of Theorem 2 is deferred to Appendix C.

**Assumption 3.** *$H(\theta)$ is convex.*

**Assumption 4.** *There exists a fixed $k$ such that $\|\theta_t - \theta^*\| \leq k\|\theta_0 - \theta^*\|$ holds for $t = 0, 1, \ldots, T-1$.*

**Theorem 2.** *Suppose that Assumptions 1-4 hold and $\alpha < \frac{1}{3}$. Taking $\eta = \frac{1}{L_H}$, Algorithm 2 satisfies*

$$H(\theta_T) - H(\theta^*) \leq \max\left\{ \frac{4L_H D^2}{(1 - (1+r)C_\alpha^2)T}, \ \sqrt{\frac{2(1+1/r)}{1 - (1+r)C_\alpha^2}}D\sigma + \frac{(1+1/r)\sigma^2}{2L_H} \right\} \tag{4}$$

*where $D = k\|\theta_0 - \theta^*\|$, $C_\alpha = \frac{2\alpha}{1-\beta}$ and $0 < r < \left(\frac{1-\beta}{2\alpha}\right)^2 - 1$, which are the same as in Theorem 1.*

**Strongly convex losses.** Finally, we assume strong convexity on the objective function as in Assumption 5, in which case we derive Theorem 3 that guarantees convergence of our algorithm to the optimal model $\theta^*$. The proof of Theorem 3 is deferred to Appendix D.

**Assumption 5.** *$H(\theta)$ is $\lambda_H$-strongly convex.*

**Theorem 3.** *Suppose that Assumptions 1-2 and 5 hold, and $\alpha < \frac{1}{1+2L_H/\lambda_H} < \frac{1}{3}$. Taking $\eta = \frac{2}{L_H+\lambda_H}$, Algorithm 2 satisfies (with $C_\alpha = \frac{2\alpha}{1-\beta}$)*

$$\|\theta_T - \theta^*\| \leq \left(\frac{2L_H C_\alpha + L_H - \lambda_H}{L_H + \lambda_H}\right)^T \|\theta_0 - \theta^*\| + \frac{\sigma}{\lambda_H - L_H C_\alpha}. \tag{5}$$

**Observations.** According to Theorems 1-3, Algorithm 2 is capable of achieving a degree of convergence under varying assumptions regarding the model's convexity. Meanwhile, we can clearly identify the effects of Byzantine percentage $\alpha$: a larger $\alpha$ (i.e., a larger $C_\alpha$) not only decreases convergence speed, but also increases convergence error. In contrast, the $\sigma$ in (2) only affects the convergence error and has no impact on the convergence rate.

### 4.3 Discussion

**The tightness of (1) in Lemma 1.** Since all three convergence theorems are derived based upon Lemma 1, it is important to examine the tightness of (1). In practice, one can almost always find a way to upper-bound $\|G - \nabla H\|$, normally through some mathematical tricks and engineering approximations. However, convergence guarantees based upon loose bounds may be less effective or even risk losing their meaningfulness altogether. In this regard, we use a one-dimensional toy problem to demonstrate the tightness of (1).

Given $\nabla H_1 = 4$, $\nabla H_2 = 5$, $\nabla H_3 = 6$, and $\nabla H = 5$. Now suppose that one of the three users is Byzantine, say user 2, who can send the server an arbitrary value $g_2$ instead of the honest gradient $\nabla H_2 = 5$. To implement NBS, we impose that $\beta = \alpha = \frac{1}{3}$. We can verify that (1) always holds, with the RHS equal to 6 and the LHS never reaching 6 no matter what value $g_2$ takes. However, the LHS can infinitely approach 6 if $g_2$ is selected judiciously. For example, for the set of gradients $g_1 = 4$, $g_2 = -5.98$, $g_3 = 6$, the LHS is equal to 5.99. In this case, the Byzantine gradient $g_2$ is carefully chosen such that it is able to bypass the screening, and meanwhile exerts maximum adversarial perturbation on the output. Readers can verify that (1) also holds when user 1 or user 3 is Byzantine. This example shows that (1) precisely measures the worst-case deviation by Byzantine users and that it cannot be further tightened.

**The advantages of NBS over its counterparts.** Aside from NBS, most of the other robust aggregation measures are robust-mean-based [7, 8, 9, 10], with the goal of computing a robust mean

gradient for every single iteration. Given the neat convergence guarantees resulted from NBS, a natural question arises: "*Can any of those robust-mean-based methods lead to similar convergence results?*" It turns out that the answer to this question might be negative, due to three major theoretical advantages of NBS over its counterparts, which make the latter not only require over-stringent assumptions but also produce less informative bounds.

First, unlike robust-mean-based methods, NBS does not require the assumption that local data are iid. As explained in Section 4.1, bounding the noise term $\|G - \nabla H\|$ is crucial for convergence, where $G$ represents the output of robust aggregation, whether it is NBS or one of its counterparts. For the latter, the iid assumption is indispensable since it ensures that the honest gradients are iid samples of the true global gradient $\nabla H$. Without this assumption, $\|G - \nabla H\|$ cannot be upper-bounded theoretically.

Second, unlike its counterparts, NBS does not require local gradients to follow a certain distribution, which often lacks proper justification in practice. For robust-mean-based methods, this distributional assumption determines the level to which honest local gradients are concentrated. Without this assumption, $\|G - \nabla H\|$ also cannot be upper-bounded theoretically. In contrast, in order to bound $\|G - \nabla H\|$, NBS only requires Assumption 1, which always holds as explained previously and does not have the justification issue.

Third (and most importantly), NBS's bound on $\|G - \nabla H\|$ is tighter, therefore more useful, than that of its counterparts. To reach convergence, the noise term $\|G - \nabla H\|$ has to be bounded for every iteration, suggesting that the distributional assumption needs to be made repeatedly for robust-mean-based methods. Although the distribution of honest gradients is supposed to become more and more concentrated as the model makes steady progress towards the optimum, such dynamic distributional change is unknown to us and there is no way to accurately incorporate this piece of information into the distributional assumption at each iteration. Therefore, one has little choice but to compromise by imposing the same distributional assumption on local gradients for all iterations, which inevitably leads to a loose bound on $\|G - \nabla H\|$. For example, the authors of [9] assume that the partial derivatives of the objective follow the same sub-exponential distribution for all iterations. As a result, they can only bound $\|G - \nabla H\|$ as a constant $\Delta$ irrespective of the iteration index (see Theorem 8 and Theorem 11 therein). Clearly, the inequality $\|G - \nabla H\| \leq \Delta$ is not tight since it completely ignores the ever-concentrating effect of honest gradients as training proceeds. In contrast, NBS leads to (1), i.e., $\|G - \nabla H\| \leq C_\alpha \|\nabla H\| + \sigma$, which accurately captures the supposed shrinkage of $\|G - \nabla H\|$ as $\|\nabla H\|$ decreases.

To summarize, NBS enjoys the above three distinctive advantages over its robust-mean-based counterparts in theoretically upper-bounding $\|G - \nabla H\|$, making it suitable for us to derive meaningful convergence guarantees.

**The effects of non-iid data.** Although our convergence analysis applies to both iid and non-iid settings, the heterogeneity of local data does have an impact on the convergence results. This is reflected in (2) where the distance between any local gradient and the true global gradient is upper-bounded by $\sigma$ ($\max_{1 \leq i \leq m} \|\nabla H_i - \nabla H\| \leq \sigma$), which is determined by data heterogeneity. In the most homogeneous case where each user has the same data, $\sigma$ is equal to 0. As data heterogeneity increases, $\sigma$ should also increase accordingly. Through the adoption of such a universal upper bound $\sigma$, we are able to derive meaningful convergence guarantees without the normally required iid assumption. This analytical approach, i.e., making assumptions in the spirit of (2) and imposing a universal upper bound to eliminate local gradients' differences, might serve as a pathway for future works to bypass the non-iid issue theoretically in the convergence analysis.

## 5 Simulation

### 5.1 Experimental setup

**Dataset and allocation.** In this section, we empirically evaluate the performance of our algorithm for a classification task using the logistic regression model on the Spambase dataset [21]. We assign $\frac{2}{3}$ of the 4601 total samples for training and the other $\frac{1}{3}$ for testing. We evenly split the training data among $m = 20$ nodes, with 8 of them exclusively holding spams (labelled as 1) and the remaining 12 exclusively holding non-spams (labelled as 0). Each node has the same number of data points $n = 153$. Although in theory there are numerous ways to create a non-iid setting, splitting data

according to their labels seems to be the common practice. In the Byzantine setting, we pick $\alpha m = 4$ nodes as Byzantine (2 from each group). Upon the completion of training, we use the percentage of correct classifications on the test set as the performance metric of model accuracy.

**Objective and fairness metric.** To achieve fairness, we adopt $q$-FFL [16] where the local objective takes the form of $H_i(\theta) = \frac{1}{q+1}[F_i(\theta)]^{q+1}$ with $F_i(\theta) = \frac{1}{n}\sum_{j=(i-1)n+1}^{(i-1)n+n} f(\theta; x_j)$ and $f(\theta; x_j)$ being the cross-entropy loss that is common for classification tasks. As explained in Section 3.2, $q$-FFL promotes egalitarian fairness. We follow [16] to use the variance of the local classification accuracy as the metric of fairness (only the accuracies of honest users are counted for $\alpha > 0$). Note that our framework is not limited to the $q$-FFL objective and it readily suits for other fairness-oriented objectives as long as they can be decomposed in the form of $H = \frac{1}{m}[H_1 + \cdots + H_m]$.

**Byzantine models and robust benchmarks.** To generate Byzantine gradients, we experiment with four different strategies, i.e., sign-flipping attack [22], label-flipping attack [9], Inner Product Manipulation (IPM) attack (for two regimes of $\epsilon = 0.5$ and $\epsilon = 50$ respectively) [23], and "A Little Is Enough" (ALIE) attack [24]. These four types of Byzantine models are commonly considered in the literature [25], and they represent a wide range of attacks varying in aspects such as the gradient norms and adversary's knowledge. To test the effectiveness of NBS, we compare it with several widely-acknowledged robust aggregation measures Krum [7], coordinate-wise median (CM) [9] and coordinate-wise trimmed mean (CTM) [9], as benchmarks, along with the plain averaging (ERM). For Krum, CTM, and NBS, the screening/trimming parameter is chosen based on the Byzantine percentage $\alpha$. This allows these robust schemes to effectively aggregate gradient information and achieve better performance compared to schemes that do not consider $\alpha$. In practice, $\alpha$ can be set as the estimated upper bound of the Byzantine percentage, as the exact value of $\alpha$ may not always be available or reliable during the training process.

### 5.2 Evaluation

First, we seek to inspect the robustness of NBS compared with other benchmarks, without incorporating $q$-FFL for the moment. This is because reasonably high accuracy should always be the first priority of model training, without which other parallel goals such as fairness would be meaningless. In Figure 1, we plot the performance curves of model accuracy for the five considered aggregation measures (ERM, NBS, CTM, CM, Krum) under various attack regimes, using learning rate $\eta = 1$ and number of iterations $T = 300$. From Figure 1, we can see that NBS enjoys the best overall performance, due to its ability to utilize redundant information as well as its resilience to large-norm attacks (such as sign-flipping attack and IPM attack with $\epsilon = 50$ where ERM performs poorly).

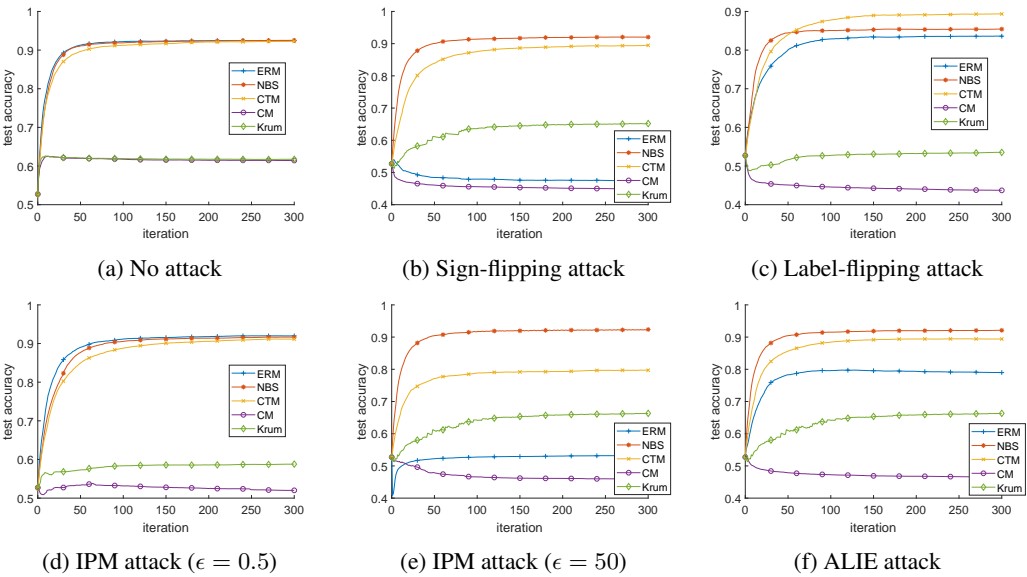

Figure 1: Model accuracy of different robust aggregation schemes under various attacks.

This result verifies that the theoretical advantages of NBS over other robust benchmarks in handling Byzantine attacks can extend to practical applications. In contrast, CM and Krum fail to converge to a functioning model even in the absence of Byzantine attacks. The only other robust scheme with comparable robustness performance to NBS is CTM, which achieves a worst-case test accuracy of 79.7% compared to 85.4% for NBS.

Next, we incorporate the $q$-FFL objective (Step 5 in Algorithm 2) into NBS and investigate how NBS, while ensuring Byzantine robustness, affects the fairness performance. In Table 1, we document both the model accuracy and the variance of local accuracies (the metric of model fairness) using different values of $q$ and screening percentage $\beta$. The learning rate $\eta$ and number of iterations $T$ for each $q$ value are carefully selected to ensure fast and stable convergence. Table 1 shows that vanilla $q$-FFL (under $\beta = 0$) achieves the intended tradeoff between utility (high accuracy) and fairness (low variance), with larger $q$ focusing more on fairness. In particular, $q = 1$ leads to a desirable tradeoff by significantly reducing the variance (from 480 to 269) while only marginally decreasing the model accuracy (from 92.5% to 91.3%). By comparison, the study in [16] reports that $q$-FFL reduces the variance of accuracies across devices by an average of 45%, with a similar decrease in overall accuracy. Therefore, we choose $q = 1$ for the $q$-FFL scheme in the remaining experiments.

Table 1: The impact of screening on the performance of H-nobs (with no attack).

|  | $\beta = 0$ | $\beta = 10\%$ | $\beta = 20\%$ | $\beta = 30\%$ | $(\eta, T)$ |
|---|---|---|---|---|---|
| $q = 0$ | 92.5% (480) | 92.7% (389) | 92.5% (396) | 92.2% (392) | (1, 300) |
| $q = 0.5$ | 92.0% (405) | 92.4% (364) | 92.3% (368) | 92.2% (381) | (0.5, 1000) |
| $q = 1$ | 91.3% (269) | 92.0% (327) | 91.9% (331) | 91.1% (349) | (0.5, 1000) |
| $q = 2$ | 86.9% (156) | 89.5% (253) | 89.0% (243) | 88.9% (260) | (0.2, 1500) |

From Table 1, we observe that increasing $\beta$ does not have an obvious effect on the model accuracy, since screening up to 30% of gradients still leaves enough redundant information in the absence of Byzantine attacks. On the other hand, the effects of screening on fairness are quite different for small $q$ and large $q$ regimes. Specifically, when $q = 0$, the variance is at its highest, which suggests a relatively wide gap on the local accuracies. In that case, screening up to 30% of gradients coincidentally suppresses some of the divisive nodes and makes the results fairer. However, when $q \geq 1$, the variance is small, in which case the screening operation begins to compromise fairness by increasing the variance, since it inevitably filters out certain fairness-enhancing gradients. This phenomenon corroborates our hypothesis in Introduction about the internal conflicts between fairness (up-weighted objective) and robustness (filtering of outliers). Despite the inevitable impaired performance on fairness compared to vanilla $q$-FFL, H-nobs (e.g., with $q = 1$ and $\beta = 20\%$) strikes a good balance between robustness and fairness, being able to defend against up to 20% of Byzantine nodes while securing some fairness gains. In addition, we observe a similar interplay between robustness and fairness in the Law School dataset [26] and the Credit Card Client dataset [27], both of which are commonly considered in the context of fairness-aware machine learning [28]. The results for these two datasets are presented in Appendix E.

Table 2: The influence of various attacks on both robustness (accuracy) and fairness (variance).

|  | $q$-FFL | $q$-FFL + NBS | $q$-FFL + CTM | $q$-FFL + CM | $q$-FFL + Krum |
|---|---|---|---|---|---|
| No attack | 91.3% (**269**) | **91.9%** (331) | 91.5% (316) | 62.0% (N/A) | 62.8% (N/A) |
| Sign-flipping | 61.1% (N/A) | **91.0%** (265) | 87.4% (**143**) | 47.3% (N/A) | 82.1% (291) |
| Label-flipping | 80.1% (150) | 81.5% (220) | **87.9%** (**61**) | 45.8% (N/A) | 57.9% (N/A) |
| IPM ($\epsilon = 0.5$) | 89.8% (**76**) | 87.1% (437) | **90.8%** (357) | 54.4% (N/A) | 57.4% (N/A) |
| IPM ($\epsilon = 50$) | 61.1% (N/A) | **90.3%** (**94**) | 71.1% (186) | 54.6% (N/A) | 88.5% (245) |
| ALIE | 76.0% (211) | **90.0%** (72) | 88.1% (**62**) | 48.4% (N/A) | 88.5% (245) |

Finally, we compare H-nobs (i.e., $q$-FFL + NBS) with the combinations of $q$-FFL and other aggregation schemes in the Byzantine setting with 4 Byzantine nodes. The robustness and fairness performance of candidate algorithms under various attacks is summarized in Table 2, using learning rate $\eta = 0.5$ and number of iterations $T = 1000$. We exclude the variance if the model accuracy falls below 65%. In Table 2, we can see that in the Byzantine-free case, both NBS and CTM effectively maintain the fairness improvements achieved by $q$-FFL, as they have a much lower variance than ERM (480), which uses the traditional training loss $F$ instead of the fair objective $H$. When Byzantine attacks are introduced, NBS demonstrates the highest level of overall robustness, achieving a worst-case test accuracy of 81.5%. In comparison, the second-best performer, CTM, reaches 71.1%. This suggesting an even wider performance gap than when considering these two schemes without integrating with $q$-FFL, which had worst-case accuracies of 85.4% and 79.7% respectively. These findings further support our choice of NBS as the aggregation method, which showcases not only theoretical but also empirical advantages over its robust-mean-based counterparts. In terms of fairness, both NBS and CTM exhibit varying performance depending on the specific attack regimes, due to the unpredictability of Byzantine gradients. As a result, no definitive conclusion can be drawn regarding the tradeoff between robustness and fairness in the presence of Byzantine attacks.

### 5.3 Summary and discussion

In this section, we first empirically illustrate the effectiveness of NBS against various Byzantine attacks and its advantage over other robust benchmarks. Then, we investigate the influence of screening on the fairness performance for H-nobs, verifying the internal conflicts between robustness and fairness. Finally, we compare H-nobs with other benchmarks under different attack regimes, demonstrating the robust performance of H-nobs and indicating that the fairness gains can vary significantly depending on the attack regimes.

We want to point out two limitations of our empirical exploration. First, our simulations are based on a fixed Byzantine percentage $\alpha = 20\%$ and a fixed level of data heterogeneity (i.e., $\sigma$ in (2)), both of which can have an impact on the convergence as demonstrated in Theorems 1-3. However, as $\alpha$ or $\sigma$ increases, overcoming Byzantine failure might become an intractable task due to the insufficient redundancy among the honest users. We leave the exploration on the robustness/fairness tradeoff under different $\alpha/\sigma$ regimes for future work. Second, our evaluation of fairness is conducted based on the specific $q$-FFL objective under the egalitarian fairness notion. Whether or not our empirical results apply to other fairness notions or objectives is an open question and deserves further investigation in the future.

## 6   Conclusion

In this paper, we recognize three important but underexplored questions related to fair and robust distributed learning. We address these questions by proposing a new framework H-nobs that achieves joint fairness and robustness. The convergence of the proposed algorithm, even under the challenging non-iid settings, is theoretically guaranteed for different types of learning models. We empirically verify the effectiveness of our algorithm under various attack regimes, and we also take the initiative to investigate the influence of robustness-oriented screening on the performance of fairness.

## Acknowledgments and Disclosure of Funding

This work was partly supported by the NSF grants No.1939553, No.2003211, and No.2231209.

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

# Appendices

## A    Proof of Lemma 1

For $G = \frac{1}{|\mathcal{U}|}\sum_{i \in \mathcal{U}} g_i$ with $\mathcal{U} = \{(1), \ldots, ((1-\beta)m)\}$ and any specific vector $S$, we have

$$
\begin{aligned}
\|G - S\| &= \left\| \frac{1}{|\mathcal{U}|} \sum_{i \in \mathcal{U}} g_i - S \right\| \\
&= \left\| \frac{1}{|\mathcal{U}|} \sum_{i \in \mathcal{U}} (g_i - S) \right\| \\
&= \frac{1}{|\mathcal{U}|} \left\| \sum_{i \in \mathcal{U} \cap \mathcal{M}} (g_i - S) + \sum_{i \in \mathcal{U} \cap \mathcal{B}} (g_i - S) \right\| \\
&\leq \frac{1}{|\mathcal{U}|} \left( \sum_{i \in \mathcal{U} \cap \mathcal{M}} \|g_i - S\| + \sum_{i \in \mathcal{U} \cap \mathcal{B}} \|g_i - S\| \right).
\end{aligned}
$$

For $i \in \mathcal{U} \cap \mathcal{M}$, $\|g_i - S\| \leq \Delta$ (we define $\Delta = \max_{i \in \mathcal{M}} \|g_i - S\|$).

For $i \in \mathcal{U} \cap \mathcal{B}$, we bound $\|g_i - S\|$ as follows:

$$
\begin{aligned}
\|g_i - S\| &\leq \|g_i\| + \|S\| \\
&\leq \|g_{((1-\beta)m)}\| + \|S\| \\
&\leq \|g_{((1-\alpha)m)}\| + \|S\| \\
&\leq \max_{i \in \mathcal{M}} \|g_i\| + \|S\| \\
&= \max_{i \in \mathcal{M}} \|g_i - S + S\| + \|S\| \\
&\leq \max_{i \in \mathcal{M}} \|g_i - S\| + 2\|S\| \\
&= \Delta + 2\|S\|.
\end{aligned}
$$

Combining the above results, we have

$$
\begin{aligned}
\|G - S\| &\leq \frac{1}{|\mathcal{U}|} \Big( |\mathcal{U} \cap \mathcal{M}| \cdot \Delta + |\mathcal{U} \cap \mathcal{B}| \cdot (\Delta + 2\|S\|) \Big) \\
&= \frac{1}{|\mathcal{U}|} \Big( |\mathcal{U}| \cdot \Delta + 2|\mathcal{U} \cap \mathcal{B}| \cdot \|S\| \Big) \\
&= \Delta + \frac{2|\mathcal{U} \cap \mathcal{B}|}{|\mathcal{U}|} \|S\| \\
&\leq \Delta + \frac{2\alpha}{1-\beta} \|S\|
\end{aligned}
\tag{6}
$$

which is exactly the conclusion in Lemma 1 after we replace $S$ with $\nabla H$. Note that in (6), the last inequality only holds on condition that $|\mathcal{B}| \leq |\mathcal{U}|$, i.e., $\alpha \leq 1 - \beta$, which, combined with $\beta \geq \alpha$, suggests that $\alpha \leq \frac{1}{2}$.

# B  Proof of Theorem 1

According to Assumption 2, $H(\theta)$ is $L_H$-smooth. According to the property of smoothness, we have

$$
\begin{aligned}
H(\theta_{t+1}) &\leq H(\theta_t) + \langle \nabla H(\theta_t), \theta_{t+1} - \theta_t \rangle + \frac{L_H}{2}\|\theta_{t+1} - \theta_t\|^2 \\
&= H(\theta_t) - \eta \langle \nabla H(\theta_t), G(\theta_t) \rangle + \frac{L_H}{2}\eta^2 \|G(\theta_t)\|^2 \\
&= H(\theta_t) - \frac{1}{L_H}\langle \nabla H(\theta_t), G(\theta_t) - \nabla H(\theta_t) + \nabla H(\theta_t) \rangle + \frac{1}{2L_H}\|G(\theta_t) - \nabla H(\theta_t) + \nabla H(\theta_t)\|^2 \\
&= H(\theta_t) - \frac{1}{L_H}\|\nabla H(\theta_t)\|^2 + \frac{1}{2L_H}\Big(\|\nabla H(\theta_t)\|^2 + \|G(\theta_t) - \nabla H(\theta_t)\|^2\Big) \\
&= H(\theta_t) - \frac{1}{2L_H}\|\nabla H(\theta_t)\|^2 + \frac{1}{2L_H}\|G(\theta_t) - \nabla H(\theta_t)\|^2
\end{aligned}
\tag{7}
$$

where the first equality follows from $\theta_{t+1} = \theta_t - \eta \cdot G(\theta_t)$ and the second equality follows from $\eta = \frac{1}{L_H}$. Note that the derivation of (7) is a common trick in analyzing the convergence of smooth functions which shifts the burden of proving convergence into the relatively easy task of quantifying $\|G(\theta_t) - \nabla H(\theta_t)\|$.

Combining Lemma 1 and Assumption 1, we have $\|G(\theta_t) - \nabla H(\theta_t)\| \leq C_\alpha \|\nabla H(\theta_t)\| + \sigma$, which leads to

$$
\begin{aligned}
\|G(\theta_t) - \nabla H(\theta_t)\|^2 &\leq C_\alpha^2 \|\nabla H(\theta_t)\|^2 + 2C_\alpha \|\nabla H(\theta_t)\|\sigma + \sigma^2 \\
&\leq (1+r)C_\alpha^2 \|\nabla H(\theta_t)\|^2 + (1 + 1/r)\sigma^2
\end{aligned}
\tag{8}
$$

for any $r > 0$.

Combining (7) and (8), we have

$$
H(\theta_{t+1}) \leq H(\theta_t) - \frac{1 - (1+r)C_\alpha^2}{2L_H}\|\nabla H(\theta_t)\|^2 + \frac{1 + 1/r}{2L_H}\sigma^2
\tag{9}
$$

which is equivalent to

$$
\|\nabla H(\theta_t)\|^2 \leq \frac{2L_H}{1 - (1+r)C_\alpha^2}\big[H(\theta_t) - H(\theta_{t+1})\big] + \frac{1 + 1/r}{1 - (1+r)C_\alpha^2}\sigma^2.
\tag{10}
$$

Summing up (10) for $t = 0, 1, \ldots, T-1$ before divided by $T$ gives

$$
\begin{aligned}
\frac{1}{T}\sum_{t=0}^{T-1}\|\nabla H(\theta_t)\|^2 &\leq \frac{2L_H}{(1 - (1+r)C_\alpha^2)T}\big[H(\theta_0) - H(\theta_T)\big] + \frac{1 + 1/r}{1 - (1+r)C_\alpha^2}\sigma^2 \\
&\leq \frac{2L_H}{(1 - (1+r)C_\alpha^2)T}\big[H(\theta_0) - H(\theta^*)\big] + \frac{1 + 1/r}{1 - (1+r)C_\alpha^2}\sigma^2
\end{aligned}
\tag{11}
$$

which is exactly the conclusion in Theorem 1.

Note that the transition from (9) to (10) only stands under the condition that $1 - (1+r)C_\alpha^2 > 0$, which constrains $r$ to the less than $\frac{1}{C_\alpha^2} - 1$. On the other hand, $r > 0$, which requires $C_\alpha = \frac{2\alpha}{1-\beta} < 1$, i.e., $2\alpha + \beta < 1$. Since $\beta \geq \alpha$, we can conclude that (11) holds if and only if $\alpha < \frac{1}{3}$ and $0 < r < \left(\frac{1-\beta}{2\alpha}\right)^2 - 1$.

## C  Proof of Theorem 2

According to Assumption 3, the convexity of $H(\theta)$ suggests that $H(\theta^*) \geq H(\theta_t) + \langle \nabla H(\theta_t), \theta^* - \theta_t \rangle$, which leads to

$$
\begin{aligned}
H(\theta_t) - H(\theta^*) &\leq \langle \nabla H(\theta_t), \theta_t - \theta^* \rangle \\
&\leq \|\nabla H(\theta_t)\| \cdot \|\theta_t - \theta^*\| \\
&\leq D\|\nabla H(\theta_t)\|
\end{aligned}
\tag{12}
$$

in which $D = k\|\theta_0 - \theta^*\|$. The third inequality of (12) follows from Assumption 4. As a result, we obtain (13) as a key property for the subsequent analysis.

$$
\|\nabla H(\theta_t)\| \geq \frac{1}{D}\big[H(\theta_t) - H(\theta^*)\big]
\tag{13}
$$

Since Theorem 2 keeps all the assumptions made in Theorem 1, all the intermediate steps in the proof of Theorem 1 also apply here. In this regard, we borrow (9), i.e.,

$$
H(\theta_{t+1}) - H(\theta_t) \leq -A\|\nabla H(\theta_t)\|^2 + B
\tag{14}
$$

in which we define $A = \frac{1-(1+r)C_\alpha^2}{2L_H}$ and $B = \frac{(1+1/r)\sigma^2}{2L_H}$ for convenience.

Next, we consider two cases in regard to the relationship between $A\|\nabla H(\theta_t)\|^2$ and $B$.

**Case 1.** Suppose for all $0 \leq t \leq T-1$, it holds that $B \leq \frac{A}{2}\|\nabla H(\theta_t)\|^2$. In this case, we have

$$
H(\theta_{t+1}) - H(\theta_t) \leq -\frac{A}{2}\|\nabla H(\theta_t)\|^2.
\tag{15}
$$

Combining (13) and (15) gives

$$
\big[H(\theta_t) - H(\theta^*)\big]^2 \leq \frac{2D^2}{A}\Big(\big[H(\theta_t) - H(\theta^*)\big] - \big[H(\theta_{t+1}) - H(\theta^*)\big]\Big)
\tag{16}
$$

which, after divided by $\big[H(\theta_t) - H(\theta^*)\big]\big[H(\theta_{t+1}) - H(\theta^*)\big]$ on both sides, leads to

$$
\frac{H(\theta_t) - H(\theta^*)}{H(\theta_{t+1}) - H(\theta^*)} \leq \frac{2D^2}{A}\Big(\frac{1}{H(\theta_{t+1}) - H(\theta^*)} - \frac{1}{H(\theta_t) - H(\theta^*)}\Big).
\tag{17}
$$

According to (15), $H(\theta_{t+1}) \leq H(\theta_t)$. Therefore, $\frac{H(\theta_t)-H(\theta^*)}{H(\theta_{t+1})-H(\theta^*)} \geq 1$, and (17) can be simplified as

$$
\frac{1}{H(\theta_{t+1}) - H(\theta^*)} - \frac{1}{H(\theta_t) - H(\theta^*)} \geq \frac{A}{2D^2}.
\tag{18}
$$

Summing up (18) for $t = 0, 1, \ldots, T-1$ gives

$$
\begin{aligned}
\frac{1}{H(\theta_T) - H(\theta^*)} &\geq \frac{AT}{2D^2} + \frac{1}{H(\theta_0) - H(\theta^*)} \\
&\geq \frac{AT}{2D^2}
\end{aligned}
\tag{19}
$$

which leads to

$$
H(\theta_T) - H(\theta^*) \leq \frac{2D^2}{AT}.
\tag{20}
$$

**Case 2.** Suppose $\exists\, t_0 \in \{0, 1, \ldots, T-1\}$, such that $B > \frac{A}{2}\|\nabla H(\theta_{t_0})\|^2$. In this case, we have

$$
\|\nabla H(\theta_{t_0})\| < \sqrt{\frac{2B}{A}}.
\tag{21}
$$

Combining (13) and (21) gives

$$
H(\theta_{t_0}) - H(\theta^*) < D\sqrt{\frac{2B}{A}}.
\tag{22}
$$

Next we show by contradiction that for all $t \geq t_0$, it holds that

$$H(\theta_t) - H(\theta^*) \leq D\sqrt{\frac{2B}{A}} + B. \tag{23}$$

Suppose that there exists $t_1 \geq t_0$ such that

$$H(\theta_{t_1}) - H(\theta^*) > D\sqrt{\frac{2B}{A}} + B. \tag{24}$$

According to (14), we have

$$\begin{aligned} H(\theta_{t_1}) - H(\theta_{t_1-1}) &\leq -A\|\nabla H(\theta_{t_1-1})\|^2 + B \\ &\leq B. \end{aligned} \tag{25}$$

Combining (24) and (25) gives

$$H(\theta_{t_1-1}) - H(\theta^*) > D\sqrt{\frac{2B}{A}}. \tag{26}$$

Combining (26) and (13) gives

$$\|\nabla H(\theta_{t_1-1})\| > \sqrt{\frac{2B}{A}}. \tag{27}$$

Plugging (27) into (14), we obtain $H(\theta_{t_1-1}) \geq H(\theta_{t_1}) + B$, which suggests that (24) also holds with $t_1$ replaced by $t_1 - 1$. By the same token, we can conclude that (24) should hold with $t_1$ replaced by all $t \leq t_1$. This is in clear contradiction with the incident of $t = t_0$ as shown in (22). Therefore, (23) is valid for all $t \geq t_0$ as stated.

Finally, combining the results of Case 1 (20) and Case 2 (23), we achieve that

$$\begin{aligned} H(\theta_T) - H(\theta^*) &\leq \max\left\{ \frac{2D^2}{AT}, D\sqrt{\frac{2B}{A}} + B \right\} \\ &= \max\left\{ \frac{4L_H D^2}{(1 - (1+r)C_\alpha^2)T}, \sqrt{\frac{2(1+1/r)}{1 - (1+r)C_\alpha^2}} D\sigma + \frac{(1+1/r)\sigma^2}{2L_H} \right\} \end{aligned} \tag{28}$$

which completes the proof of Theorem 2.

# D  Proof of Theorem 3

According to Assumption 2, $H(\theta)$ is $L_H$-smooth, and according to Assumption 5, $H(\theta)$ is $\lambda_H$-strongly convex. In convex optimization theory, it is well known that smooth and strongly convex functions enjoy linear convergence rate with gradient descent. Here we will first establish and then use such a property with a specific convergence factor. We start with the following equality

$$\|\theta_t - \eta\nabla H(\theta_t) - \theta^*\|^2 = \|\theta_t - \theta^*\|^2 - 2\eta\langle\nabla H(\theta_t), \theta_t - \theta^*\rangle + \eta^2\|\nabla H(\theta_t)\|^2. \quad (29)$$

According to the co-coercivity of smooth and strongly convex function, we have

$$\langle\nabla H(\theta_t) - \nabla H(\theta^*), \theta_t - \theta^*\rangle \geq \frac{1}{L_H + \lambda_H}\|\nabla H(\theta_t) - \nabla H(\theta^*)\|^2 + \frac{L_H\lambda_H}{L_H + \lambda_H}\|\theta_t - \theta^*\|^2. \quad (30)$$

Since $\theta^*$ is the global minimizer of $H(\theta)$ and therefore $\nabla H(\theta^*) = 0$, (30) reduces to

$$\langle\nabla H(\theta_t), \theta_t - \theta^*\rangle \geq \frac{1}{L_H + \lambda_H}\|\nabla H(\theta_t)\|^2 + \frac{L_H\lambda_H}{L_H + \lambda_H}\|\theta_t - \theta^*\|^2. \quad (31)$$

Plugging (31) into (29), we have

$$\|\theta_t - \eta\nabla H(\theta_t) - \theta^*\|^2 \leq \left(1 - 2\eta\frac{L_H\lambda_H}{L_H + \lambda_H}\right)\|\theta_t - \theta^*\|^2 + \left(\eta^2 - \frac{2\eta}{L_H + \lambda_H}\right)\|\nabla H(\theta_t)\|^2. \quad (32)$$

In order to eliminate the last term in (32), we take $\eta = \frac{2}{L_H + \lambda_H}$ and simplify (32) as

$$\|\theta_t - \eta\nabla H(\theta_t) - \theta^*\|^2 \leq \left(1 - \frac{4L_H\lambda_H}{(L_H + \lambda_H)^2}\right)\|\theta_t - \theta^*\|^2 \quad (33)$$

which is the same as

$$\|\theta_t - \eta\nabla H(\theta_t) - \theta^*\| \leq \frac{L_H - \lambda_H}{L_H + \lambda_H}\|\theta_t - \theta^*\| \quad (34)$$

which verifies linear convergence with a factor of $\frac{L_H - \lambda_H}{L_H + \lambda_H}$. Note that (34) holds on condition that $\eta = \frac{2}{L_H + \lambda_H}$.

Next, we try to evaluate the single-step progress made by our algorithm as follows:

$$\begin{aligned}
\|\theta_{t+1} - \theta^*\| &= \|\theta_t - \eta G(\theta_t) - \theta^*\| \\
&= \|\theta_t - \eta\nabla H(\theta_t) - \theta^* + \eta[\nabla H(\theta_t) - G(\theta_t)]\| \\
&\leq \|\theta_t - \eta\nabla H(\theta_t) - \theta^*\| + \eta\|\nabla H(\theta_t) - G(\theta_t)\| \\
&\leq \frac{L_H - \lambda_H}{L_H + \lambda_H}\|\theta_t - \theta^*\| + \frac{2}{L_H + \lambda_H}\|G(\theta_t) - \nabla H(\theta_t)\| \\
&\leq \frac{L_H - \lambda_H}{L_H + \lambda_H}\|\theta_t - \theta^*\| + \frac{2C_\alpha}{L_H + \lambda_H}\|\nabla H(\theta_t)\| + \frac{2\sigma}{L_H + \lambda_H}
\end{aligned} \quad (35)$$

in which the second inequality follows from (34) by taking $\eta = \frac{2}{L_H + \lambda_H}$, and the third inequality follows from Lemma 1 and Assumption 1.

According to the properties of $H(\theta)$ being $L_H$-smooth, we have

$$\frac{1}{2L_H}\|\nabla H(\theta_t)\|^2 \leq H(\theta_t) - H(\theta^*) \leq \frac{L_H}{2}\|\theta_t - \theta^*\|^2$$

which leads to

$$\|\nabla H(\theta_t)\| \leq L_H\|\theta_t - \theta^*\|. \quad (36)$$

Plugging (36) into (35), we have

$$\|\theta_{t+1} - \theta^*\| \leq \frac{2L_H C_\alpha + L_H - \lambda_H}{L_H + \lambda_H}\|\theta_t - \theta^*\| + \frac{2\sigma}{L_H + \lambda_H}. \quad (37)$$

By iterating (37) we obtain

$$\|\theta_T - \theta^*\| \leq \left(\frac{2L_H C_\alpha + L_H - \lambda_H}{L_H + \lambda_H}\right)^T\|\theta_0 - \theta^*\| + \frac{\sigma}{\lambda_H - L_H C_\alpha} \quad (38)$$

which is exactly the conclusion in Theorem 3.

Note that the transition from (37) to (38) only stands under the condition that $\frac{2L_H C_\alpha + L_H - \lambda_H}{L_H + \lambda_H} < 1$, which requires $C_\alpha = \frac{2\alpha}{1-\beta} < \frac{\lambda_H}{L_H}$, i.e., $2\alpha\frac{L_H}{\lambda_H} + \beta < 1$. Since $\beta \geq \alpha$, we can conclude that (38) holds if and only if $\alpha < \frac{1}{1 + 2L_H/\lambda_H}$.

# E   The interplay between robustness and fairness in two additional datasets

Table 3: The performance of H-nobs on the Law School dataset (with no attack).

|  | $\beta = 0$ | $\beta = 10\%$ | $\beta = 20\%$ | $\beta = 30\%$ | $(\eta, T)$ |
|---|---|---|---|---|---|
| $q = 0$ | 90.7% (559) | 90.6% (523) | 90.6% (512) | 90.6% (507) | (0.1, 500) |
| $q = 0.5$ | 89.8% (483) | 90.3% (478) | 90.3% (470) | 90.4% (465) | (0.05, 1000) |
| $q = 1$ | 87.3% (353) | 89.9% (449) | 90.2% (451) | 90.2% (448) | (0.02, 2000) |
| $q = 2$ | 66.2% (228) | 73.3% (428) | 74.0% (447) | 73.9% (445) | (0.005, 5000) |

Table 4: The performance of H-nobs on the Credit Card Client dataset (with no attack).

|  | $\beta = 0$ | $\beta = 10\%$ | $\beta = 20\%$ | $\beta = 30\%$ | $(\eta, T)$ |
|---|---|---|---|---|---|
| $q = 0$ | 80.7% (1195) | 80.5% (1143) | 80.6% (1181) | 80.6% (1148) | (0.2, 500) |
| $q = 0.5$ | 81.0% (1123) | 80.3% (1081) | 80.3% (1140) | 80.4% (1105) | (0.1, 2000) |
| $q = 1$ | 81.0% (987) | 80.1% (1006) | 79.8% (1058) | 80.0% (1035) | (0.1, 2000) |
| $q = 2$ | 78.9% (869) | 79.1% (887) | 79.0% (930) | 79.3% (966) | (0.05, 5000) |

For both the Law School dataset and the Credit Card Client dataset, we use $\frac{2}{3}$ as the training/testing split ratio and divide the training data into two groups based on the male/female feature (instead of the label), which is a commonly considered attribute in fairness research. Similar to the previous setup, we distribute the grouped training data across 20 nodes in a balanced manner. The performance of H-nobs on these two datasets is detailed in Table 3 and Table 4, respectively.

Tables 3 and 4 exhibit a pattern similar to that in Table 1: when the variance is high (under small $q$ regimes), increasing $\beta$ does not impair fairness; however, when $q$-FFL achieves significant fairness gains (under large $q$ regimes), fairness is compromised by the screening operation. Note that here we only consider the effects of screening, not in combination with attacks. When Byzantine attacks are at play, Table 2 demonstrates satisfying robustness performance of H-nobs, although the fairness gains achieved by $q$-FFL can vary considerably depending on the attack regimes.

