# OpenReview forum: "H-nobs: Achieving Certified Fairness and Robustness in Distributed Learning on Heterogeneous Datasets"
_NeurIPS.cc/2023/Conference — NeurIPS 2023 poster_

### Official Review · Reviewer_9BqC · 2023-07-06

**Soundness:** 3 good
**Presentation:** 2 fair
**Contribution:** 2 fair
**Rating:** 4
**Confidence:** 3

**Summary:**

This paper introduces a fair and robust algorithm in distributed learning setup, which combines q-FFL [8] (or the one by [6]) with the norm-based screening robust method. Three convergence results w.r.t. the algorithm are developed for non-convex, convex and strongly convex models. Experimental results are presented on the Spambase dataset.

**Strengths:**

1. The paper addresses an important issue dealing with both fairness and robustness aspects in distributed learning.

2. It derives three convergence theorems based on Lemma 1. The tightness of the bound in Lemma 1 is discussed via an illustrated example.

3. The effectiveness of the considered algorithm and the relationship between fairness and robustness are explored via experiments.


**Weaknesses:**

1. The developed algorithm looks a simple combination of the prior works: ([6],[8]) for fairness objectives and [4] for gaining insights for NBS.

2. The convergence theorems are derived under a certain regime, alpha <1/3. Also it  is not quite clear as to how the theorems explain the advantage of NBS over its counterparts. In addition, it would be more informative if the implication and meaning of the statements in the theorems are elaborated.

3. It considers only one fairness notion, egalitarian fairness, under accuracy disparity. Any implications to other fairness notions and those beyond accuracy disparity?

4. Oracle baselines (centralized fair and robust algorithms prevalent in the literat) could be considered in experiments. Also more extensive experiments (beyond only Spambase dataset) are preferred to be included.


**Questions:**

Please see the Weaknesses in the above.

**Limitations:**

Please see the Weaknesses in the above.

---

> ### Author Rebuttal · Authors · 2023-08-10
>
> **Comment:** *The developed algorithm looks a simple combination of the prior works: ([6],[8]) for fairness objectives and [4] for gaining insights for NBS.*
>
> **Response:**
> Thank you for your comment. While it is true that the key elements of our framework, NBS and fairness objectives, both come from prior works, we want to emphasize that no previous endeavor has incorporated the two (or any other robust aggregation scheme with fairness objectives). The seeming simplicity of such combination belies its actual complexity, which we expound upon in Section 4.3. Accordingly, we illustrate how NBS can be employed to address the major theoretical challenges, which might attract other researchers and inspire them to come up with better ideas and results. In addition, we take the lead in investigating the internal conflicts between fairness and robustness. Our preliminary results validate our initial hypotheses, potentially inspiring greater engagement within the community to invest more effort and gain a deeper understanding of the topic. For more details, please refer to the **[Significance of This Work]** section in the global response.
>
> **Comment:** *The convergence theorems are derived under a certain regime $\alpha<\frac{1}{3}$. Also it is not quite clear as to how the theorems explain the advantage of NBS over its counterparts. In addition, it would be more informative if the implication and meaning of the statements in the theorems are elaborated.*
>
> **Response:**
> Thank you for your comment. Although the theoretical convergence of our algorithm can only be achieved under no more than 33\% of Byzantine users, we want to point out that this would have little practical impact. In a realistic non-iid setting, if the actual Byzantine percentage exceeds 20\%, achieving robustness might become an insurmountable task due to the lack of available redundancy (see Fig. 1(e)-(h) in the attached PDF). As in Lines 228-230, we demonstrate that the typical and prevalent robust-mean-based methods not only require stringent/impractical assumptions but also lead to uninformative/loose bounds, making them unfavorable compared to NBS. For example, as one of very few works that provide convergence guarantees for robust-mean-based methods, [17] requires the iid assumption as well as the sub-exponential distributional assumption on the local gradients at all iterations (see Assumption 6 therein). In return, it can only bound $\|G-\nabla H\|$ as a universal constant, which is directly applied to the developed convergence theorems, making the resulted inequality highly loose and potentially impractical. In contrast, our convergence results are grounded in a closely bounded $\|G-\nabla H\|$, offering a more informative and realistic perspective. Moreover, our results achieve this without necessitating the assumptions of iid-ness and specific distributions on the gradients. Finally, Lines 203-206 demonstrate the respective influence of the Byzantine percentage $\alpha$ and the data heterogeneity level $\sigma$ on the convergence under multiple levels of assumptions on the model convexity.
>
> **Comment:** *It considers only one fairness notion, egalitarian fairness, under accuracy disparity. Any implications to other fairness notions and those beyond accuracy disparity?*
>
> **Response:**
> Thank you for your question. We believe it is important to provide clarification on this matter, which is addressed in the global response. Please see the **[Fairness Evaluation on $q$-FFL]** section therein.
>
> **Comment:** *Oracle baselines (centralized fair and robust algorithms prevalent in the literature) could be considered in experiments. Also more extensive experiments (beyond only Spambase dataset) are preferred to be included.*
>
> **Response:**
> As far as we understand, centralized algorithms are not applicable in the distributed setting unless they can be made decomposable/parallelizable through some means. Therefore, centralized fair and robust algorithms may not be comparable with our framework. Also, we have done additional empirical studies but did not include them into the manuscript. Please see the **[Additional Empirical Settings]** section in the global response for more details.

---

> > ### Comment · Area_Chair_FyVa · 2023-08-18
> >
> > Dear Reviewer 9bqc,
> >
> > We would appreciate if you could acknowledge and/or respond to the authors' rebuttal.
> >
> > Thank you,
> >
> > AC

---

### Official Review · Reviewer_waox · 2023-07-06

**Soundness:** 4 excellent
**Presentation:** 4 excellent
**Contribution:** 3 good
**Rating:** 6
**Confidence:** 3

**Summary:**

This paper proposes a protocol to address both fairness and Byzantine robustness in federated learning, and specifically i) the objective function is designed to take fairness into account, and ii) the algorithm takes an average of gradients with large norms and reduces the influences of large gradients. In addition to theoretical analysis, another important part of this paper is the discussion of these problems: 1) what makes achieving both fairness and robustness difficult; 2) to what extent does fairness harm robustness.

**Strengths:**

1. The paper thoroughly discusses the deep problem of fairness and robustness; such discussion in Section 1 provides insights on the principal aspects on some unexplored questions.

2. The theoretical analysis is extensive, with multiple levels of assumptions (e.g. non-convexity vs convexity vs strong convexity) and corresponding guarantees.

3. Section 4.3 provides convincing defense on the NBS protocol.

**Weaknesses:**

1. I’m not sure why having a large norm (at least possibly) implies maliciousness? I’m not an expert in Byzantine robustness, but if this is a result in this area, I’d certainly prefer to have a reference or at least a brief discussion.

2. Even if the discussion in Section 4.3 clearly explains the advantages of the protocol, but claiming it as a “unique” protocol with those advantages (non-IID, no distributions on gradients, theoretical guarantees) may lack support.

**Questions:**

I do not have additional questions other than those in the weaknesses section.

**Limitations:**

As mentioned in the weaknesses section, one may not be convinced that the NBS protocol is the uniquely suitable one with those advantages. It may be reasonable to mention this or alternatively provides more explicit support for that claim.

---

> ### Author Rebuttal · Authors · 2023-08-10
>
> **Comment:** *I’m not sure why having a large norm (at least possibly) implies maliciousness? I’m not an expert in Byzantine robustness, but if this is a result in this area, I’d certainly prefer to have a reference or at least a brief discussion.*
>
> **Response:**
> Thank you for your question. Let's first consider the one-dimensional case, where the averaging operation is sensitive to outliers (compare $\frac{2+2+2}{3}=2$ and $\frac{2+2+200}{3}=68$). In order to mitigate the influence of outliers, robust measures will discard the largest values when doing aggregation to avoid the worst case as in $\frac{2+2+200}{3}$. Similarly, in high-dimensional cases, large-norm gradients are deemed as outliers that can potentially cause the most harm and therefore need to be removed.
>
> **Comment:** *Even if the discussion in Section 4.3 clearly explains the advantages of the protocol, but claiming it as a “unique” protocol with those advantages (non-IID, no distributions on gradients, theoretical guarantees) may lack support. Therefore, one may not be convinced that the NBS protocol is the uniquely suitable one with those advantages. It may be reasonable to mention this or alternatively provides more explicit support for that claim.*
>
> **Response:**
> Thank you for this keen observation. Indeed, having certain advantages does not necessarily make a scheme unique and it is quite possible to discover other schemes with similar traits in the future. We appreciate your constructive suggestion and will change the wording in the revised version to avoid overstatement.

---

> > ### Comment · Reviewer_waox · 2023-08-16
> >
> > I thank the author for explaining the relationship between norm value and maliciousness. I have decided to keep my original rating.

---

### Official Review · Reviewer_fgN6 · 2023-07-06

**Soundness:** 3 good
**Presentation:** 3 good
**Contribution:** 2 fair
**Rating:** 6
**Confidence:** 3

**Summary:**

The paper explores robustness to byzantine attacks/failures as well as fairness in a federated learning setting.
The main focus of the paper is on a heterogenous user setting, i.e. data is non-iid between users. Particularly, the paper considers egalitarian fairness, i.e. uniform performance for all users.
To address this setting the paper combines (previously proposed) fairness-promoting losses with norm-based screening (NBS) for Byzantine robustness.
In a theoretical analysis, the authors show convergence (under suitable assumptions) where convergence speed depends on the number of Byzantine users.
In an empirical evaluation, the proposed method compares favorably in terms of accuracy, even in the presence of malicious users, and fairness (as measured by variance).

**Strengths:**

- Byzantine robustness aspect is well done: theoretical grounding, empirical evaluation (fairness less so, see below).
- Well written and easy to follow.
- Good empirical results.
- Intersection of fairness and robustness is an interesting and important topic. As a non-expert (in federated learning and the particular notion of fairness) this appears as a reasonable contribution to the field.

**Weaknesses:**

- With regards to novelty and contribution: Both NBS and fairness-promoting losses are not novel. While there is definitely value in the combination, "framework ... containing them" (L64) seem to be a bit of a stretch. The theoretical analysis of NBS though indeed seems novel.
- Minor presentation issues.

See questions below.

**Questions:**

- Can you clarify the contributions of the paper and comment on the novelty of the different aspects of the proposed approach?
- Can you clarify whether the $\eta$ in Line 143 (hyper-parameter of the fairness-aware loss) and $\eta$ in Algorithm 2 Line 11 (learning rate) are indeed different?


EDIT POST REBUTTAL: As my main question as sufficiently addressed I have increased my score.

**Limitations:**

A short explicit paragraph on limitations is given and limitations are discussed throughout the text.
This seems sufficient for the work at hand.

---

> ### Author Rebuttal · Authors · 2023-08-10
>
> **Comment:** *With regards to novelty and contribution: Both NBS and fairness-promoting losses are not novel. Can you clarify the contributions of the paper and comment on the novelty of the different aspects of the proposed approach?*
>
> **Response:**
> Thank you for your insightful question. Although both NBS and fairness-promoting losses are not novel individually, no prior work has combined the two, or any other robust aggregation scheme with a fair objective. This combination might appear straightforward but is, in fact, highly challenging and non-trivial, as discussed in Section 4.3. The primary innovation of our work lies in establishing the theoretical foundation for attaining certified fairness and robustness within challenging non-iid scenarios. We achieve this by showcasing how NBS can be employed to address the formidable theoretical challenges, which might attract other researchers and inspire them to come up with better ideas and results. In addition, we take the lead in investigating the internal conflicts between fairness and robustness. Our preliminary results validate our initial hypotheses, potentially inspiring greater engagement within the community to invest more effort and gain a deeper understanding of the topic. In summary, the major contributions of our work are listed as follows:
> - We recognize the major challenges in obtaining joint fairness and robustness in distributed learning by identifying three important but largely unnoticed and unaddressed questions. Through this work, we have addressed these questions to varying degrees.
> - We propose H-nobs, the first algorithm that takes the (obvious but challenging) route of incorporating robust aggregation into existing fair objectives to achieve fair and robust distributed learning.
> - We achieve certified fairness and robustness for H-nobs even under the challenging non-iid scenarios, by establishing theoretical convergence guarantees with multiple levels of assumptions on the model convexity. Also, we clearly identify the major technical challenges that may hinder this goal, and illustrate how those challenges can be overcome by NBS through its theoretical robust property.
> - We empirically demonstrate the robustness and fairness benefits of H-nobs and take the initiative to investigate the empirical conflicts between fairness and robustness under a specific metric and objective of fairness.
>
> **Comment:** *Can you clarify whether the $\eta$ in Line 143 (hyper-parameter of the fairness-aware loss) and $\eta$ in Algorithm 2 Line 11 (learning rate) are indeed different?*
>
> **Response:**
> Yes, they are indeed different. We should have used $\mu$ for the hyper-parameter in Line 143 instead of $\eta$, which denotes the learning rate. Thank you for this keen observation.

---

> > ### Comment · Reviewer_fgN6 · 2023-08-11
> > **Reply**
> >
> > I thank the authors for their clarification.
> > I still encourage the authors to extend their discussion from here to the next revision of their paper as to me (as a reader mostly working on adjacent fields -- see confidence 3) it was hard to assess the novelty and contribution of this work while reading.
> >
> > As this adequately addresses my main weakness I have updated my score. At the current moment I do not have further questions.

---

### Official Review · Reviewer_hrXW · 2023-07-07

**Soundness:** 1 poor
**Presentation:** 2 fair
**Contribution:** 2 fair
**Rating:** 5
**Confidence:** 4

**Summary:**

This paper acknowledges and tackles the challenges associated with fair and robust distributed learning. It addresses these challenges by introducing the H-nobs framework and provides convergence guarantees under different convexity assumptions. The proposed algorithm is then evaluated through experiments under various attack regimes to demonstrate its effectiveness.

**Strengths:**

* Addressing the combinatorial issue of achieving fairness across clients and robustness in federated learning holds significant importance.
* Convergence guarantees under various convexity assumptions are provided, along with the respective discussion.
* The paper addresses novel questions regarding robustness and fairness trade-offs in FL settings.

**Weaknesses:**

* My main concern is the experimental evaluation. In particular, the paper considers just one dataset, examines a single value of $\alpha$, one data heterogeneity setting. A single setting cannot provide enough evidence to thoroughly evaluate the applicability and utility of the proposed approach with confidence.

* The reproducibility of the presented results is limited due to the insufficient information provided about the experiments, such as the learning rate, the value of $q$, and other relevant details.

* The number of Byzantine users $\alpha$ must be known a priori and is fixed throughout the training process.


*  The work assumes that the data is balanced across clients, which may be unrealistic in federated learning settings. Although the paper briefly acknowledges that imbalancedness can be supported (line 126), there is limited practical information provided on how this can be effectively implemented or realized.

**Questions:**

* Can H-nobs support minimax fairness notions such as AFL or DRO, as stated in line 143? Given that NBS removes the gradients with large norms, my understanding is that picking such fairness notions might result in a model with low fairness or even no utility.
    If this is indeed the case, this ans similar statements require revision.

* Is there a principled way to define $\alpha$? In my view, there should be at least an experimental ablation study showcasing how $\alpha$ impacts the performance of the proposed approach.

* What is the value of $q$ for the q-FFL in the experiments? Other relevant implementation details about hyper-paramenters are missing etc.,

Minor:

* references across the paper are not hyperlinked
* line 128: does the percentage $\alpha \in [0,1)$?

**Limitations:**

The authors could further enhance their discussion on the limitations of the proposed work. While they acknowledge the limitations related to fixed data heterogeneity and Byzantine users, there are additional experimental constraints that should be acknowledged. Additionally, it would be valuable for the authors to address any potential conflicts between the proposed framework and certain fairness definitions that may require further attention. For more details, please refer to the weaknesses and questions highlighted earlier.

---

> ### Author Rebuttal · Authors · 2023-08-10
>
> **Comment:** *My main concern is the experimental evaluation. In particular, the paper considers just one dataset, examines a single value of $\alpha$, one data heterogeneity setting. A single setting cannot provide enough evidence to thoroughly evaluate the applicability and utility of the proposed approach with confidence.*
>
> **Response:**
> Thank you for your insightful comment. We believe that thoroughly addressing this concern is very important. Please refer to the **[Fairness Evaluation on $q$-FFL]** and **[Additional Empirical Settings]** sections in the global response, where we provide sufficient clarification on this matter.
>
> **Comment:** *The reproducibility of the presented results is limited due to the insufficient information provided about the experiments, such as the learning rate, the value of $q$, and other relevant details.*
>
> **Response:**
> Thank you for pointing out this issue. Here is a detailed breakdown of our experiment executions.
> - **Data split.** As in Lines 275-278, we evenly split the training data among 20 nodes, 8 of which only hold spams (labelled as 1) and the other 12 only hold non-spams (labelled as 0). In the Byzantine setting, we pick 4 nodes as Byzantine (2 from each group). We set $\alpha m=4$ because our previous work shows that NBS can tolerate up to 4 Byzantine nodes while guaranteeing 80\% test accuracy.
> - **Hyper-parameters.** For Figure 1, we set the learning rate $\eta=1$ and number of iterations $T=300$ for all the tested cases. For all the experiments in Table 1 and Table 2, we set $q=1$, $\eta=0.5$ and $T=1000$. We use a smaller learning rate for the latter case because it is easier to diverge due to the up-weighting effect of $q$-FFL. Throughout, the five (or six) tested algorithms all start from the same randomly-generated initial model, and each experiment is repeated 10 times using different random initialization to obtain the averaged results.
>
> **Comment:** *The number of Byzantine users $\alpha$ must be known a priori and is fixed throughout the training process.*
>
> **Response:**
> Thank you for your comment. We believe that having to know/fix $\alpha$ beforehand may not necessarily be a weakness, since knowing $\alpha$ allows the scheme to flexibly aggregate the available information and achieve better performance. To see this, let's compare two coordinate-wise schemes: CTM, which requires $\alpha$, and CM, which does not. From Figure 1 in the manuscript, we notice that CTM consistently outperforms CM under all cases by a large margin. This is because CTM utilizes a flexible combination of filtering and aggregating based on $\alpha$. In practice, one can set $\alpha$ as the estimated upper-bound of Byzantine percentage, say 20\%. Note that in a practical non-iid setting, if the actual Byzantine percentage exceeds 20\%, then achieving robustness might become an insurmountable task due to the lack of available redundancy (see Fig.1 (e)-(h) in the attached PDF).
>
> **Comment:** *The work assumes that the data is balanced across clients, which may be unrealistic in federated learning settings. Although the paper briefly acknowledges that imbalancedness can be supported (line 126), there is limited practical information provided on how this can be effectively implemented or realized.*
>
> **Response:**
> Thank you for your question. To demonstrate unbalanced data allocation, let's consider a simplified case with two users, holding 3 and 7 samples respectively. Using the objective in Line 143, the local training losses on these two users are $H_1(\theta)=\frac{1}{3}\sum_{j=1}^{3}[f(\theta;x_j)-\mu]^2_+$ and $H_2(\theta)=\frac{1}{7}\sum_{j=4}^{10}[f(\theta;x_j)-\mu]^2_+$ respectively, which reflect a sense of fairness and can be incorporated into our framework.
>
>
> **Comment:** *Can H-nobs support minimax fairness notions such as AFL or DRO ...*
>
> **Response:**
> To address this concern, we want to point out that H-nobs only removes large-norm gradients on an iterative basis, which means that the honest users whose gradients are discarded in certain iterations would always have a chance to catch up in later iterations. Also, within a single iteration, even if the honest gradients with the largest norms are removed, not all is lost because the surviving honest gradients still carry the goal of fairness, which is reflected in the expression of up-weighted objectives in both Line 143 and 144. On the other hand, discarding the gradients with the largest norms and siding with the majority of all gradients is the defining characteristic of all Byzantine-robust approaches, not just NBS. Although such a practice would inevitably cause conflicts with fairness, it may not necessarily lead to an unfair model, as illustrated in our empirical exploration.
>
> **Comment:** *Is there a principled way to define $\alpha$? In my view, there should be at least an experimental ablation study showcasing how $\alpha$ impacts the performance of the proposed approach.*
>
> **Response:**
> As explained in a previous response, one can set $\alpha$ as the estimated upper-bound of Byzantine percentage, which normally should not exceed 20\%. Also, we have done such an empirical study as suggested but did not include it into the original manuscript. Please see the **[Additional Empirical Settings]** section in the global response.
>
> **Comment:** *What is the value of $q$ for the q-FFL in the experiments? Other relevant implementation details about hyper-parameters are missing etc.*
>
> **Response:**
> We set $q=1$ throughout, which leads to decent fairness gains in our setting (see Table 1). The $q$-FFL paper [8] also uses $q=1$ for some empirical cases (see Table 1 therein). Note that we do not present results with different $q$ values because our focus is to address the conflicts between fairness and robustness, instead of finding the best $q$ value for $q$-FFL. For more implementation details, please refer to our response to your second comment regarding reproducibility.

---

> > ### Comment · Reviewer_hrXW · 2023-08-15
> >
> > I thank the authors for their response. Let me provide further clarifications regarding the primary concerns on empirical evaluation that have not been addressed.
> >
> > > Insufficient empirical evaluation
> >
> > This paper wants to draw accurate conclusions about the fairness robustness conflicts based on the results on a single dataset, a single $q$ value (for instantiating the fairness objective) and a single $\alpha$ value, using logistic regression.
> >
> > Let me elaborate more on what the problem is with that setting:
> >
> >
> > 1. **Empirical conclusions are not generalizable:** In order for the experimental findings to be generalizable, more than just a single dataset must be examined. I am interested to understand why and how one can assert that the findings from a single dataset are representative and have broader generalizability.
> >
> > 2. **On picking a single value of $q$:** $q$ defines the amount of fairness considered, with larger $q$ resulting in more fairness (i.e., lower variance) and smaller $q$ focuses more on the utility/accuracy of the model. You state that $q=1$ yields decent fairness gains and that this is shown in Table 1. Why? Did you examine other $q$ values to confidently assert that $q=1$ achieves that?
> > Given the flexibility of $q$-FFL approach, there will not be an optimal $q$ value as the authors state, but different values should be examined to understand the achievable fairness-robustness trade-offs.
> > I also note that the rationale provided for selecting $q=1$ based on [8]'s usage of the same value in some experiments is insufficient, particularly since (1) [8]'s results indicate the $q$ is dataset-dependent (except if q=0 or q extremely large), and (2) [8] does not analyze/consider the same dataset as the current work.
> >
> >
> > 3. **Ablation study about $\alpha$ and [Additional Empirical Settings] section:** Can the authors please clarify whether on the attached results they use H-nobs algorithm rather than just NBS? My comment about the empirical evaluation is about the proposed approach (H-nobs).
> >
> > About my points 2 \&3: I strongly believe that examining the different conflicts between fairness and robustness requires studying whether the connection between the number of byzantine users and fairness level. Hence, it is reasonable to examine the interplay and impact of $\alpha$ and the focus put on fairness for various values of $q$ to accurately characterize conflicts between the two, (i.e., examine more settings for table 2).
> >
> >
> > **I would be happy to revise my score if the authors address the concerns above.**
> >
> > Minor comments:
> >
> > > on knowing $\alpha$ a prior.
> >
> > I understand the utility of fixing/knowing the $\alpha$ beforehand, however, the authors should recognise and add to their discussion that having access to such information prior to the training process might not be feasible in many scenarios.
> >
> > > Reproducibility
> >
> > I thank the authors for providing information on the experimental setup. The missing hyper-parameter details should be added to the paper to make this work reproducible.
> >
> > > on H-nobs supporting minimax fairness notions and your response in **[Fairness Evaluation on q-FFL]**
> >
> > NBS might remove updates from clients that are contributing to the worst-case fairness, especially if those clients have gradients that deviate from the norm due to their unique characteristics. This could potentially hinder the achievement of minimax fairness. I encourage the authors to state this scenario and its impact in the manuscript. Especially, given that both metrics you consider in lines 143 and 144 can recover minimax fairness. It would be also ideal to include experiments on that (i.e., by using a sufficiently large $q$ value to recover AFL).

---

> > > ### Author Response · Authors · 2023-08-18
> > >
> > > Thank you for clarifying your question. When drafting our manuscript, we experimented with different values of $q$ and selected the one ($q=1$) with a good accuracy/fairness tradeoff. As per your suggestion, we have rerun our algorithm using different $q$ and screening percentage $\alpha$ (without Byzantine attacks), under the setup that is consistent with our paper. The results are presented in TABLE A.
> > >
> > > **TABLE A** Evaluation on the Spambase Dataset
> > > |         | $\alpha=0$  | $\alpha=0.1$ | $\alpha=0.2$ | $\alpha=0.3$ | $\eta$, $T$  |
> > > |---------|-------------|--------------|--------------|--------------|--------------|
> > > | $q=0$   | 92.4% (481) | 92.7% (389)  | 92.5% (394)  | 92.4% (382)  | (1, 300)     |
> > > | $q=0.5$ | 92.2% (414) | 92.5% (364)  | 92.3% (364)  | 92.3% (379)  | (0.5, 1000)  |
> > > | $q=1$   | 91.5% (268) | 92.0% (335)  | 91.9% (337)  | 90.7% (363)  | (0.5, 1000)  |
> > > | $q=2$   | 87.3% (152) | 89.5% (266)  | 89.0% (257)  | 88.9% (264)  | (0.2, 1500)  |
> > >
> > > From TABLE A, we can see that under $q=0$, the variance is at its highest (481), suggesting a relatively wide gap on the local performances. In that case, screening up to 30% of gradients coincidentally suppresses some of the divisive nodes and makes the results fairer, while maintaining high accuracy. However, as $q$ reaches 1 and above, the variance is small, in which case NBS starts to compromise fairness by ignoring certain fairness-enhancing gradients, a phenomenon presented in our paper. In addition, we found that in the iid setting, the variance under ERM is very small (less than 10), and adjusting neither $q$ nor $\alpha$ has an obvious effect on model fairness or accuracy.
> > >
> > > To address your concern on the generalizability of our results, we repeated the above simulations using logistic regression on the Law School dataset and the Credit Card Client dataset, both of which are commonly considered for fairness-aware machine learning [R1]. The results are presented in TABLE B and TABLE C respectively.
> > >
> > > **TABLE B** Evaluation on the Law School Dataset
> > > |         | $\alpha=0$  | $\alpha=0.1$ | $\alpha=0.2$ | $\alpha=0.3$ | $\eta$, $T$   |
> > > |---------|-------------|--------------|--------------|--------------|---------------|
> > > | $q=0$   | 90.7% (559) | 90.6% (523)  | 90.6% (512)  | 90.6% (507)  | (0.1, 500)    |
> > > | $q=0.5$ | 89.8% (483) | 90.3% (478)  | 90.3% (470)  | 90.4% (465)  | (0.05, 1000)  |
> > > | $q=1$   | 87.3% (353) | 89.9% (449)  | 90.2% (451)  | 90.2% (448)  | (0.02, 2000)  |
> > > | $q=2$   | 66.2% (228) | 73.3% (428)  | 74.0% (447)  | 73.9% (445)  | (0.005, 5000)   |
> > >
> > > **TABLE C** Evaluation on the Credit Card Client Dataset
> > > |         | $\alpha=0$   | $\alpha=0.1$ | $\alpha=0.2$ | $\alpha=0.3$ | $\eta$, $T$   |
> > > |---------|--------------|--------------|--------------|--------------|---------------|
> > > | $q=0$   | 80.7% (1195) | 80.5% (1143) | 80.6% (1181) | 80.6% (1148) | (0.2, 500)    |
> > > | $q=0.5$ | 81.0% (1123) | 80.3% (1081) | 80.3% (1140) | 80.4% (1105) | (0.1, 2000)   |
> > > | $q=1$   | 81.0% (987)  | 80.1% (1006) | 79.8% (1058) | 80.0% (1035) | (0.1, 2000)   |
> > > | $q=2$   | 78.9% (869)  | 79.1% (887)  | 79.0% (930)  | 79.3% (966)  | (0.05, 5000)  |
> > >
> > > In TABLE B and TABLE C, we observe a similar pattern: increasing $\alpha$ does not impair fairness if the variance is high (under small $q$), but the trend is reversed when $q$-FFL achieves significant fairness gains. Note that here we only consider the effects of screening, not in combination with attacks. When Byzantine attacks are at play, our paper shows the robustness gains of NBS in comparison to other screening schemes. Meanwhile, the fairness gains achieved by $q$-FFL can vary considerably depending on the attack regimes.
> > >
> > > **Additional experimental details**
> > >
> > > For all three datasets, we use $\frac{2}{3}$ as the training/testing split ratio and evenly allocate the training data to $m=20$ nodes in a non-iid manner. For the latter two datasets, we use the male/female feature (a commonly considered attribute) to divide two groups of nodes. The learning rate $\eta$ and number of iterations $T$ for each experiment are appended in the tables and are carefully selected to ensure fast and stable convergence. All the results here are the average of 5 trials with different random initialization.
> > >
> > > **To address your minor comments**
> > >
> > > Thank you for your constructive suggestions. In the revision, we will strengthen the discussions on the pros/cons of fixing $\alpha$, the experimental details, and the potential effects of our algorithm on minimax fairness. Note that here we do not present the results with a very large $q$ because it makes the convergence volatile and does not always achieve the intended effects. (For instance, in TABLE II, using $q=3$ leads to both the reduction of accuracy and the increase of variance.)
> > >
> > > **Reference**
> > >
> > > [R1] Le Quy, et al. “A survey on datasets for fairness‐aware machine learning.” *Wiley Interdisciplinary Reviews: Data Mining and Knowledge Discovery* 12.3 (2022): e1452.

---

> > > > ### Comment · Reviewer_hrXW · 2023-08-19
> > > >
> > > > I thank the authors for their response. I have revised my score accordingly. Currently, I do not have any additional questions.

---

### Author Rebuttal · Authors · 2023-08-10

We thank all the reviewers for their valuable time and helpful comments. Here we first want to highlight two key aspects of our work and then address two major concerns raised by the reviewers. In particular, we note that a reject recommendation was made mainly based on the extensiveness of our simulations, not on the technical merits of the proposed work. We provide detailed explanation on our evaluation to demonstrate that the evaluation is convincing, and thus does not warrant a reject decision.

### [Significance of This Work]

- First, while deriving convergence guarantees, our work clearly exposes the decisive technical challenges thwarting this process, (i.e., stringent assumptions and loose bounds), and how those challenges can be overcome using NBS. Such revelations are the first of their kind, and may inspire future researchers to explore the commonalities of all qualifying schemes and propose other solutions beyond NBS.
- Second, our proposed framework encompasses a broad category of fair objectives (i.e., decomposable), thus not limited to $q$-FFL. Unfortunately, there are not many such objectives in the current literature, since the goal of fairness has mostly been considered under the traditional (centralized) setting where all the data are accessible at a single node. As a result, the developed objectives often are not directly decomposable. However, this situation may soon change for the better as the distributed setting gains popularity these days, and we observe a trend of more centralized algorithms being customized into the distributed setting. For example, the objective in Line 143 was originally adapted from a centralized DRO objective, and then made decomposable using the duality transformation. The significance of our theoretical framework lies in that, whenever new decomposable fairness-aware objectives are developed in the future, our theoretical guarantees would still be applicable.

### [Fairness Evaluation on $q$-FFL]

In Lines 354-357, we comment that our empirical evaluation on fairness is focused on $q$-FFL, while the evaluation of other fairness metrics is left for future work. Both Reviewer hrXW and Reviewer 9BqC raised a concern that such evaluation does not appear to be general enough. Following our previous argument, we want to reiterate the current shortage of decomposable fairness-aware objectives. In fact, we only have two such objectives in the distributed setting (in Lines 143-144). We decide to stick with only one in order to streamline the presentation and avoid potential confusions. $q$-FFL is our choice because a) it is associated with egalitarian fairness, which is a well-defined fairness metric; b) it has better empirical performance than the alternative. Based upon the $q$-FFL objective, our work makes two important observations:
- Table 1 shows that under the Byzantine-free setting, the fairness gains introduced by $q$-FFL are unanimously reduced by the integration of robust measures. This result corroborates our analysis in Introduction about the internal conflicts between fairness (up-weighted objective) and robustness (filtering of outliers).
- Table 2 shows that under the Byzantine-prone setting, the negative correlation between fairness and model accuracy no longer holds due to the unpredictability of Byzantine gradients. Specifically, the fairness gains of each candidate algorithm can vary significantly depending on the attack regimes, and there appears to be no definitive conclusions on the tradeoff between robustness and fairness.

Since the objective in Line 143 also utilizes the idea of up-weighting, we can expect very similar observations if we incorporate another set of experiments based on this alternative objective (and its own fairness metric), which may appear to be cumbersome/redundant for our presentation and also confusing due to the new fairness metric. In short, our evaluation of $q$-FFL might appear to be limiting, but is convincing enough to reveal the intricacy in addressing both fairness and robustness issues for general distributed learning applications. We note that Reviewer hrXW, while recognizing the significance and novelty of our work, made a reject decision citing limited evaluation. We hope that our justifications on the evaluation portion have sufficiently addressed such concerns.

### [Additional Empirical Settings]

Reviewer hrXW pointed out the lack of an experiment on the impact of $\alpha$. Such an empirical study has actually been done by another paper of ours (referred to as Paper A), which focuses on addressing both distributional shift and Byzantine failure. Paper A thoroughly compares NBS with other benchmarks under various circumstances and it finds that NBS functions well under different $\alpha$ and data distribution. In the attached PDF, we incorporate two tables and one figure that validate the effectiveness of NBS:
- TABLE I compares the test accuracy of different aggregation rules under $\alpha m=6$ Byzantine nodes (out of $m=20$) in the iid setting, and shows that all robust measures perform well due to the high level of homogeneity among the local gradients.
- TABLE II documents the same experiment in the non-iid setting and shows that no scheme can outperform all the other schemes across the board, but NBS clearly enjoys the best overall performance.
- Fig. 1 displays the performance curves under varying number of Byzantine nodes in both iid and non-iid settings. In the non-iid setting, NBS can tolerate up to 4 Byzantine nodes while guaranteeing 80\% test accuracy in all cases, the best among all candidates.

Note that we did not include too many of these results into the manuscript because our focus is the interaction between fairness and robustness. Nevertheless, we hope that these results offer convincing arguments on the usefulness of NBS and H-nobs. Also, we can send Paper A to the committee if requested, which is currently under submission to a journal.

---

### Decision · Program_Chairs · 2023-09-21

**Decision:**

Accept (poster)

**Comment:**

This work explores tensions between fairness and robustness in distributed learning, and proposes an approach for solving a fair objective while enabling robustness to malicious updates. Reviewers agreed that this is an important problem to consider and appreciated the theoretical exploration of the topic, including rigorously studying the tension between fairness/robustness. However, as noted by multiple reviewers (and partially addressed during updated experiments in the rebuttal phase), the paper could be strengthened significantly by revising/updating the empirical section to include a broader set of datasets and more thorough comparison with baselines and existing methods.